# Maternal intrahepatic cholestasis of pregnancy and neurodevelopmental conditions in offspring: A population-based cohort study of 2 million Swedish children

**Shuyun Chen**[1‡]*, **Viktor H. Ahlqvist**[1‡]*, **Hugo Sjöqvist**[1], **Olof Stephansson**[2,3], **Cecilia Magnusson**[1,4], **Christina Dalman**[1,4], **Håkan Karlsson**[5], **Brian K. Lee**[1,6,7], **Renee M. Gardner**[1]

**1** Department of Global Public Health, Karolinska Institutet, Stockholm, Sweden, **2** Clinical Epidemiology Division, Department of Medicine Solna, Karolinska Institutet, Stockholm, Sweden, **3** Department of Women's Health, Division of Obstetrics, Karolinska University Hospital, Stockholm, Sweden, **4** Centre for Epidemiology and Community Medicine, Region Stockholm, Stockholm, Sweden, **5** Department of Neuroscience, Karolinska Institutet, Stockholm, Sweden, **6** A.J. Drexel Autism Institute, Drexel University, Philadelphia, Pennsylvania, United States of America, **7** Department of Epidemiology and Biostatistics, Drexel University School of Public Health, Philadelphia, Pennsylvania, United States of America

‡ These authors share first authorship on this work.
* shuyun.chen@ki.se (SC); viktor.ahlqvist@ki.se (VHA)

**Data Availability Statement:** Swedish privacy law prohibits us from making register data publicly

## Abstract

### Background

Intrahepatic cholestasis of pregnancy (ICP) is the most common obstetric liver disorder and is associated with an increased risk of iatrogenic preterm birth and adverse infant outcomes. Hence, there are several plausible pathways through which ICP could affect offspring neurodevelopment. However, to the best of our knowledge, no studies have investigated these associations. Thus, we aimed to determine whether ICP is associated with offspring neurodevelopmental conditions.

### Methods and findings

In this Swedish register-based cohort study, we included singleton non-adopted children born in Sweden between the 1st of January 1987 and the 31st of December 2010, who were resident in Sweden >5 years, with no missing covariate information, which we followed until the 31st of December 2016. Maternal ICP diagnosis and the date of the initial diagnosis during pregnancy were obtained from the National Patient Register. Offspring diagnoses of attention deficit/hyperactivity disorder (ADHD), autism, or intellectual disability were obtained from the National Patient Register, and the dispensation of ADHD medications were obtained from the Prescribed Drug Register. Odds ratios (ORs) and 95% confidence intervals (CIs) were estimated using logistic regression while controlling for observed confounders and unobserved confounders shared among full siblings and maternal full cousins.

A total of 2,375,856 children were included in the study; 81.6% of them were of Nordic origin, and 51.4% were male. Of these, 10,378 (0.44%) were exposed to ICP. During a median

available. The data supporting our findings were used under license and ethical approval for the current study. Readers interested in obtaining microdata or replicating our study may seek similar approvals and inquire through Statistics Sweden. For further advice see: https://www.scb.se/en/services/guidance-for-researchers-and-universities/, or contact Statistics Sweden at: mikrodata@scb.se.

**Funding:** This work was supported by grants from the Swedish Research Council [grant numbers 2017–02900 (to R.M.G.) and 523–2010-1052 (to C.D.)], from StratNeuro [Strategic Research Area Neuroscience at the Karolinska Institutet) (to R.M. G.)] and from China Scholarship Council [grant numbers 201907930020 (to S.C.)]. The funders had no role in study design, data collection and analysis, decision to publish, or preparation of the manuscript.

**Competing interests:** The authors have declared that no competing interests exist.

**Abbreviations:** ADHD, attention deficit/hyperactivity disorder; BMI, body mass index; CI, confidence interval; ICP, intrahepatic cholestasis of pregnancy; NDC, neurodevelopmental conditions; OR, Odds ratio; wkGA, weeks of gestational age.

of 18 years follow-up (interquartile range 11 to 24), 143,746 (6.05%) of children were diagnosed with a neurodevelopmental condition. After adjusting for child's sex, birth year, birth month, maternal age, highest parental education level, maternal birth country, birth order, maternal psychiatric history, ICP was associated with increased odds of offspring neurodevelopmental conditions (OR 1.22, 95% CI 1.13 to 1.31), particularly among those exposed to early-onset ICP (OR 2.38, 95% CI 1.71 to 3.30) as compared to ICP diagnosed after reaching term (≥37 weeks of gestation) (OR 1.08, 95% CI 0.97 to 1.20). The findings of early-onset ICP were consistent in family-based analyses. Within-family comparisons of full maternal cousins yielded an OR of 2.99 (95% CI 1.48 to 6.04), and comparisons of full siblings showed an OR of 1.92 (95% CI 0.92 to 4.02), though the latter was less precise. The findings were consistent across specific neurodevelopmental conditions and different analytical approaches. The primary limitations of this study included its observational design, the absence of data on ICP therapeutics, and the lack of bile acid measures.

## Conclusions

In this study, we observed that exposure to ICP during gestation is associated with an increased likelihood of neurodevelopmental conditions in offspring, particularly in cases of early-onset ICP. Further studies are warranted to better understand the role of early-ICP in offspring neurodevelopment.

---

## Author summary

### Why was this study done?

- Intrahepatic cholestasis of pregnancy (ICP) is a common liver disorder during pregnancy characterized by rising bile acid levels, and it is associated with early delivery and adverse infant outcomes.

- Less is known about the long-term outcomes of children exposed to ICP. Because ICP may plausibly affect the development of the fetus either directly or via its association with adverse pregnancy outcomes, this study examined the hypothesis that ICP would be associated with an increased likelihood of neurodevelopmental conditions in children exposed during gestation.

### What did the researchers do and find?

- The study analyzed data from the Swedish registers including children born between 1987 and 2010, with follow-up for neurodevelopmental outcomes until the end of 2016. The study recorded cases of mothers diagnosed with ICP and documented the timing of these diagnoses during pregnancy using patient registries. Associations between these diagnoses and neurodevelopmental conditions, including attention deficit/hyperactivity disorder (ADHD), autism, or intellectual disability in children, were estimated using multiple analytical approaches.

- The results suggested that children exposed to ICP during pregnancy were more likely to receive diagnoses of neurodevelopmental conditions, particularly when ICP was diagnosed early in pregnancy (before 28 weeks of gestation).

- Because the associations were similar when children were compared to their maternal cousins and to their siblings, these findings do not appear to be explained by factors shared within families, such as genetics and some aspects of the early life environment, that can also influence the likelihood of neurodevelopmental conditions.

### What do these findings mean?

- Diagnosis of ICP during pregnancy, especially early in pregnancy, is associated with an increased likelihood of neurodevelopmental disorders in the children exposed during gestation.

- Because this study is observational, it is not possible to determine whether ICP is a causative factor in the development of neurodevelopmental conditions in children born to affected mothers.

- This study did not include information on bile acid concentrations among the pregnant women, and this study was conducted before treatment (using ursodeoxycholic acid) was widely used in Sweden. It will be important for future studies to consider if therapeutic modulation of bile acid levels in pregnant women affected by ICP can mitigate the associations we observe.

## Introduction

Intrahepatic cholestasis of pregnancy (ICP) is the most common obstetric liver disorder, affecting 0.5% to 2% of pregnant women [1]. ICP is characterized by pruritus (itching) and liver dysfunction with elevated serum bile acid concentration and/or liver aminotransferases [1–3]. It typically occurs in the late second or third trimester of pregnancy. Though the etiology of ICP is not completely understood, it is likely caused by a combination of genetic, hormonal, and environmental factors [1]. The main treatment for ICP is ursodeoxycholic acid, which is administered to alleviate pruritus. In severe cases, early elective delivery may be necessary, especially since ICP typically resolves rapidly after delivery [2].

Although ICP can cause significant discomfort for pregnant women, it is typically not associated with severe maternal morbidity, but may increase the risk of later hepatobiliary disease [3]. However, ICP is associated with an elevated risk of adverse outcomes for infants, including stillbirth, preterm birth, and admission to neonatal care unit [4]. Despite some of these factors, such as preterm birth, being associated with adverse neonatal outcomes, including neurodevelopmental conditions, to the best of our knowledge, there have been no previous studies examining neurodevelopmental conditions among offspring exposed to ICP during pregnancy.

We addressed this knowledge gap by examining the association between ICP in offspring neurodevelopmental conditions (i.e., attention-deficit/hyperactivity disorder [ADHD], autism, and intellectual disability) in a population-based cohort using the Swedish health and administrative registries. The consecutive structural and functional growth of the fetal brain throughout gestation [5] implies that there may be critical periods of exposure during fetal brain

development. We therefore also investigated whether the gestational week of ICP diagnosis differentially affects the likelihood of offspring neurodevelopmental conditions. Finally, capitalizing on the nationwide register-linkage of individuals across and within families, we explored the extent to which these associations could be explained by unmeasured factors shared between individuals of varying degrees of relatedness.

## Methods

The main analysis was prospectively planned (S2 Appendix), and all sensitivity analyses were developed during the research process. Reviewers requested a crude analysis, an analysis that adjusts for gestational age and an analysis with detailed imputation of maternal pre-pregnancy body mass index (BMI); these analyses were therefore not part of our original prospective plan.

### Study population

We created a population-based cohort study of mothers and offspring by linking information across nationwide registers using the unique identification number assigned to each Swedish resident at immigration or birth [6]. We included children born in Sweden between the 1st of January 1987 and the 31st of December 2010 ($n$ = 2,545,022) and linked these children to their biological mothers, fathers, and maternal grandparents using the Total Population Register [7]. We excluded children not recorded in the Medical Birth Register (covering 97% to 99% of all children born in Sweden, depending on the year of birth) [8], children who resided less than 5 years in Sweden, children who were adopted, non-singletons and children with missing covariate information (Fig 1A). Our final analytical population included 2,375,856 children born to 1,308,096 mothers, who were followed until the 31 December 2016. Ethical approval was obtained from the Stockholm regional ethical review committee, with the need for informed consent waived (DNR 2010/1185-31/5, 2016/987-32). This study is reported as per the Strengthening the Reporting of Observational Studies in Epidemiology (STROBE) guideline (S1 Appendix).

### Maternal intrahepatic cholestasis of pregnancy

Maternal ICP was identified by a recorded diagnosis in the National Patient Register (ICD-9: 646.7, ICD-10: O26.6) [9], which covers all inpatient care (and specialized outpatient care after 2001) provided in Sweden [10]. ICP is not universally screened for during pregnancy; typically, patients present with symptoms of itching. Abnormalities in bile acid concentrations and liver function tests usually emerge days or weeks later, leading to a diagnosis of ICP. We retained the first diagnosis date of ICP as a proxy for the date of onset. In the main analysis, we classified the timing of ICP according to 3 categories: before the third trimester (<28 weeks), early in the third trimester (28 to 36 weeks), and late in the third trimester (≥37 weeks).

### Offspring neurodevelopmental conditions

The primary outcome was neurodevelopmental conditions identified using the National Patient Register [11]. We defined a child as having a neurodevelopmental condition if they were ever diagnosed with ADHD (ICD-9: 314, ICD-10: F90), autism (ICD-9: 299, ICD-10: F84), or intellectual disability (ICD-9: 317–319, ICD-10: F70-F79). We also used the Prescribed Drug Register (from July 2005 and onwards), which records all dispensed prescription medications in Sweden, to identify ADHD based on the dispensation of commonly used ADHD medications [12]: methylphenidate [ATC: N06BA04] or atomoxetine [ATC: N06BA09]. We considered any neurodevelopmental condition as our primary outcome (diagnosed with at

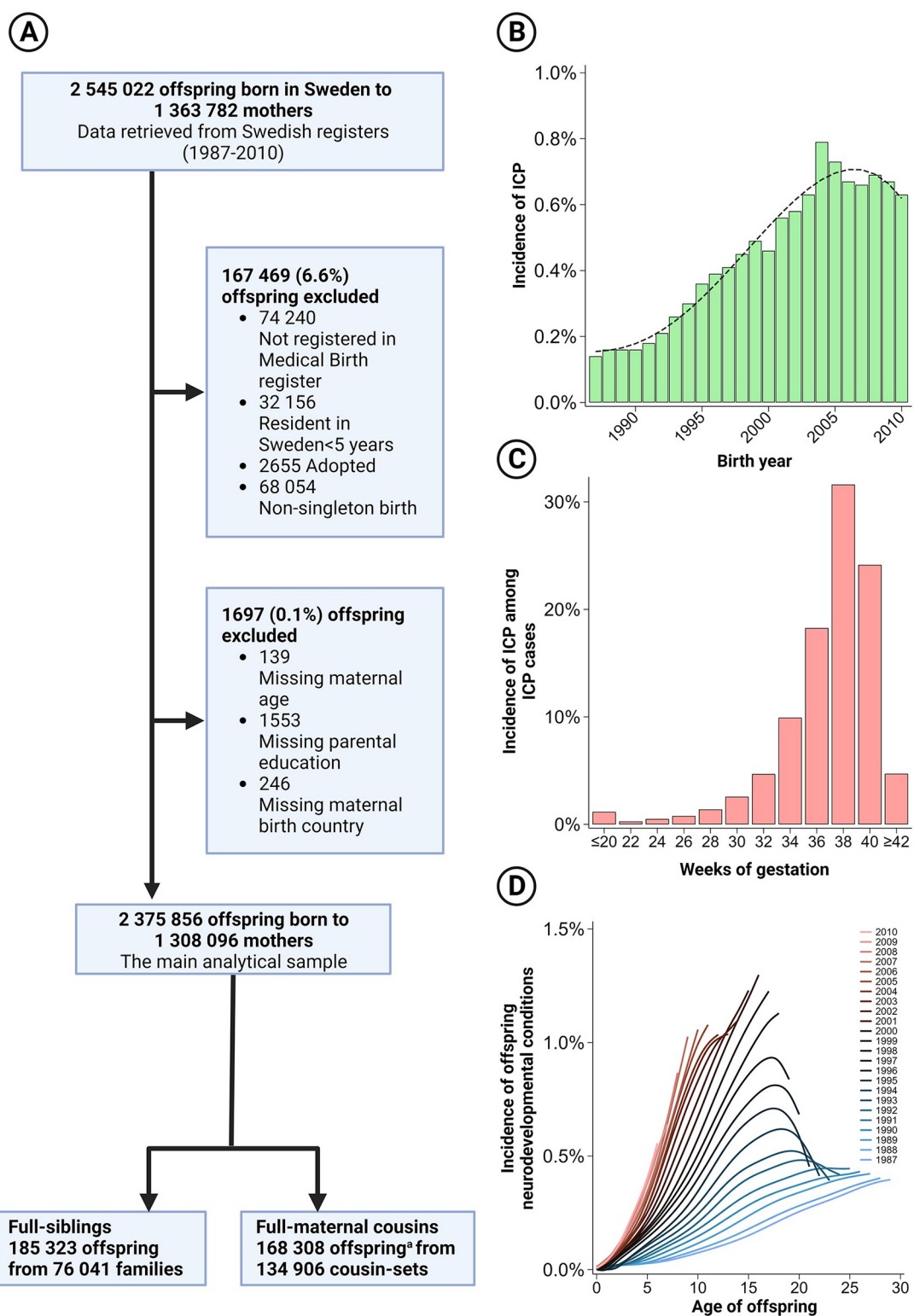

**Fig 1. Overview of the derivation of the analytical sample (A), the incidence of maternal intrahepatic cholestasis across offspring birth years (B), the gestational time of onset of intrahepatic cholestasis (C), and the age-specific incidence of offspring neurodevelopmental conditions across birth cohorts (D).**[a]As each individual can have multiple cousins, we formed 134,906 1:1 matched cousin sets, where an individual could contribute to multiple sets (i.e., be the comparative cousin to different index individuals). Fig 1D was manually calculated, and for illustrative purposes, a median spline with 50 cross-median knots was fitted to the aggregate incidences. Figure created using BioRender.com, used under license/permission.

least one of the 3 neurodevelopmental conditions). As secondary outcomes, we also separately examined ADHD, autism, and intellectual disability.

## Covariates

We collected a series of potential confounding variables (factors theorized or confirmed to have a causal impact on both the exposure and the outcome) from the Medical Birth Register, which contains detailed pregnancy and delivery information on virtually all deliveries in Sweden [13], the National Patient Register and the longitudinal integrated database for health insurance and labor market studies (LISA) [14]: the child's sex [15–18], birth year [16,17,19], birth month [1,17], birth order [19,20], maternal age [1,17,18], birth country [1,16,17], early pregnancy body mass index [17,19] (from 1992, see "Sensitivity Analyses" below) and psychiatric history [21,22], and the highest parental education level [17,19]. We also collected information on pregnancy and fetal characteristics (when available), which we considered to occur concurrently with or after onset of ICP and therefore consider as potential mediators of the relationships examined in this study: gestational diabetes mellitus, gestational hypertensive conditions, mode of delivery, labor induction, gestational age at birth, Apgar score at 5-minutes, birth weight for gestational age, neonatal asphyxia, hypoglycemia, and jaundice (S1 Table).

## Statistical analysis

To estimate the odds ratio (OR) and 95% confidence intervals (95% CI) of offspring neurodevelopmental conditions associated with maternal ICP and timing of ICP, we use logistic regression models with robust standard errors (to account for multiple children being born to the same mother). To enhance clinical interpretability, we also calculated the average absolute adjusted risk (marginal probability) of neurodevelopmental conditions based on the aforementioned models [23]. We present 3 models: one crude model without any adjustments, one model where we adjust for the child's sex and birth year, and our main model, where we further adjust for birth month, birth order, maternal age, maternal country of birth and maternal psychiatric history, and the highest parental education level. In our main analysis, we did not adjust for gestational age, as we consider it to be a mediator or a collider (S5 Fig). However, in complementary analyses not registered in the protocol, we adjusted for gestational age using 3 different approaches: as a linear term in weeks, as a categorical term (<37, 37-, 38-, 39-, 40-, 41-, 42-), and by employing restricted cubic splines with 4 knots placed at the 5th, 35th, 65th, and 95th percentile.

**Family-informed analysis.** To account for unobserved genetic and environmental confounders shared between full siblings, we performed a sibling analysis of N = 185,323 full siblings using conditional logistic regression analysis with robust standard errors. Furthermore, as full siblings are often concordant on ICP status and only discordant pairs can contribute to the familial analysis, we also performed an analysis where we matched each child to a maternal full cousin. As each child can have multiple cousins, we formed K = 134,906 matched cousin pairs (1:1 matched), where an individual could contribute to multiple pairs (i.e., be the comparative cousin to different index children). Full sibling analysis accounts for, on average, 50% of the children's genotype and maternal full cousin analysis accounts for, on average, 12.5% of the children's genotype. Similarly, we may assume that full siblings (on average) share more environmental factors than cousins. Furthermore, as ICP is a maternal phenotype, it is also relevant to note that sibling analysis accounts for 100% of the maternal genotype and maternal full cousin analysis accounts for, on average, 50% of the maternal genotype. The family-based

analysis was adjusted for all observed putative confounders which differed among relatives (e.g., not maternal education for full siblings).

**Sensitivity analyses.** We performed several sensitivity analyses. First, we replicated our primary analyses among participants with identifiable siblings and cousins to ensure that these individuals were not systematically different from that of the main cohort. Second, we replicated our primary analyses while excluding those with ICP diagnosed at delivery, as some of these ICP cases may have debuted sometime during pregnancy but were recorded as occurring at delivery. Third, we excluded all children born preterm to assess whether the association between ICP and neurodevelopmental conditions is fully explained by the effect of ICP on preterm birth. We used adjusted Wald tests to compare the results from these sensitivity analyses with the primary analysis. Fourth, we controlled for early pregnancy body mass index categories as defined by the World Health Organization among those born $\geq$ 1992, while either (i) performing complete case analysis; (ii) treating those with missing data as a separate category ("missing-as-indicator"); (iii) performing multiple imputation on categorical BMI (across 20 datasets with multinomial imputation and the exposure, outcome and covariates as auxiliary variables); or (iv) using inverse-probability weights to correct for missing patterns, since we chose not to control for it in our primary analyses due to it being poorly recorded in the Medical Birth Register (14.11% missing $\geq$ 1992). Fifth, we restricted the sample to those who were exposed to ICP and compared the odds of neurodevelopmental conditions between those diagnosed with ICP before 28 weeks, 28 to 36 weeks, and after 37 weeks of gestation (reference). Sixth, instead of categorizing the timing of ICP, we modeled the gestational week at diagnosis as a continuous variable using restricted cubic splines with 4 knots (placed at the 14, 28, 34, and 37 weeks of gestation) to relax the homogeneity assumption of the categorization. Seventh, we modeled the cumulative exposure to ICP as the number of weeks of fetal exposure to ICP (calculated from date of diagnosis to date of birth and categorized into <5, 5–8, 9–12, 13–16, and $\geq$17 weeks of exposure) and as the percentage of pregnancy exposed using restricted cubic splines models with 4 knots (placed at the 30%, 50%, 80%, and 90%).

## Results

Among the 2,375,856 children included in this study, 10,378 (0.44%) were exposed to ICP (Fig 1B), which was most commonly diagnosed in the late third trimester (*N* = 344, 0.01%, <28 weeks of gestation; *N* = 3,759, 0.16%, 28 to 36 weeks of gestation and *N* = 6,275, 0.26%, $\geq$37 weeks of gestation) (Fig 1C). Children born to mothers with ICP were more often male, first-born, born in recent years and between January and March; and more often had well-educated, older mothers, born in a Nordic country, with previous psychiatric conditions, with overweight/obesity, and had more obstetric and neonatal complications (i.e., gestational diabetes mellitus, gestational hypertensive conditions, large for gestational age, cesarean section, preterm birth, low Apgar score, neonatal asphyxia-related comorbidities, neonatal hypoglycemia, and neonatal jaundice) (Table 1).

During a median follow-up of 18 years (interquartile range 11 to 24), 143,746 (6.05%) of children were diagnosed with a neurodevelopmental condition (either ADHD, autism, or intellectual disability): 106,381 (4.48%) with ADHD, 48,363 (2.04%) with autism, and 23,933 (1.01%) with intellectual disability. These conditions increased in prevalence across birth cohorts (Fig 1D).

### Primary analyses

Children born to mothers with ICP were more likely to be diagnosed with neurodevelopmental conditions (Fig 2 and Table 2). Adjusting for potential confounders (i.e., child's sex, birth

**Table 1. Characteristics of the 2,375,856 children, born to 1,308,096 mothers between 1987 and 2010, who were included in the study, stratified by maternal ICP diagnosis.**

| | No ICP | ICP | Gestational week of ICP diagnosis | | |
| --- | --- | --- | --- | --- | --- |
| | | | <28 weeks | 28–36 weeks | ≥37 weeks |
| **Total N** | 2,365,478 | 10,378 | 344 | 3,759 | 6,275 |
| **Child's sex** | | | | | |
| Female | 1,149,882 (48.6%) | 4,791 (46.2%) | 172 (50.0%) | 1,757 (46.7%) | 2,862 (45.6%) |
| **Birth year** | | | | | |
| 1987–1992 | 668,473 (28.3%) | 1,137 (11.0%) | 34 (9.9%) | 418 (11.1%) | 685 (10.9%) |
| 1993–1998 | 567,648 (24.0%) | 2,021 (19.5%) | 47 (13.7%) | 578 (15.4%) | 1,396 (22.2%) |
| 1999–2004 | 525,904 (22.2%) | 3,122 (30.1%) | 102 (29.7%) | 1,040 (27.7%) | 1,980 (31.6%) |
| 2005–2010 | 603,453 (25.5%) | 4,098 (39.5%) | 161 (46.8%) | 1,723 (45.8%) | 2,214 (35.3%) |
| **Birth months** | | | | | |
| January–March | 604,761 (25.6%) | 2,799 (27.0%) | 78 (22.7%) | 1,035 (27.5%) | 1,686 (26.9%) |
| April–June | 632,404 (26.7%) | 2,759 (26.6%) | 93 (27.0%) | 1,007 (26.8%) | 1,659 (26.4%) |
| July–September | 606,931 (25.7%) | 2,350 (22.6%) | 96 (27.9%) | 824 (21.9%) | 1,430 (22.8%) |
| October–December | 521,382 (22.0%) | 2,470 (23.8%) | 77 (22.4%) | 893 (23.8%) | 1,500 (23.9%) |
| **Maternal age** | | | | | |
| <25 | 448,618 (19.0%) | 1,360 (13.1%) | 57 (16.6%) | 508 (13.5%) | 795 (12.7%) |
| 25–29 | 803,497 (34.0%) | 3,054 (29.4%) | 99 (28.8%) | 1,087 (28.9%) | 1,868 (29.8%) |
| 30–34 | 731,623 (30.9%) | 3,645 (35.1%) | 105 (30.5%) | 1,296 (34.5%) | 2,244 (35.8%) |
| 35–39 | 319,254 (13.5%) | 1,891 (18.2%) | 65 (18.9%) | 697 (18.5%) | 1,129 (18.0%) |
| ≥40 | 62,486 (2.6%) | 428 (4.1%) | 18 (5.2%) | 171 (4.5%) | 239 (3.8%) |
| **Highest parental education level** | | | | | |
| Primary school | 78,200 (3.3%) | 244 (2.4%) | 6 (1.7%) | 104 (2.8%) | 134 (2.1%) |
| Upper secondary school | 984,862 (41.6%) | 3,480 (33.5%) | 150 (43.6%) | 1,295 (34.5%) | 2,035 (32.4%) |
| University level | 1,302,416 (55.1%) | 6,654 (64.1%) | 188 (54.7%) | 2,360 (62.8%) | 4,106 (65.4%) |
| **Maternal birth country** | | | | | |
| Nordic | 2,035,556 (86.1%) | 9,108 (87.8%) | 279 (81.1%) | 3,236 (86.1%) | 5,593 (89.1%) |
| Europe | 100,591 (4.3%) | 329 (3.2%) | 11 (3.2%) | 141 (3.8%) | 177 (2.8%) |
| Africa | 43,686 (1.8%) | 126 (1.2%) | 13 (3.8%) | 41 (1.1%) | 72 (1.1%) |
| Asia | 154,686 (6.5%) | 582 (5.6%) | 34 (9.9%) | 246 (6.5%) | 302 (4.8%) |
| Other | 30,959 (1.3%) | 233 (2.2%) | 7 (2.0%) | 95 (2.5%) | 131 (2.1%) |
| **Maternal history of psychiatric conditions** | 119,450 (5.0%) | 801 (7.7%) | 47 (13.7%) | 363 (9.7%) | 391 (6.2%) |
| **Birth order** | | | | | |
| 1 | 1,014,257 (42.9%) | 4,677 (45.1%) | 143 (41.6%) | 1,679 (44.7%) | 2,855 (45.5%) |
| 2 | 856,290 (36.2%) | 3,636 (35.0%) | 122 (35.5%) | 1,287 (34.2%) | 2,227 (35.5%) |
| ≥3 | 494,931 (20.9%) | 2,065 (19.9%) | 79 (23.0%) | 793 (21.1%) | 1,193 (19.0%) |
| **Early-pregnancy Maternal BMI[a]** | | | | | |
| Underweight (<18.5 kg/m$^2$) | 57,543 (2.4%) | 246 (2.4%) | 9 (2.6%) | 93 (2.5%) | 144 (2.3%) |
| Normal weight (18.5–24.9 kg/m$^2$) | 1,160,457 (49.1%) | 5,469 (52.7%) | 146 (42.4%) | 1,860 (49.5%) | 3,463 (55.2%) |
| Overweight (25.0–29.9 kg/m$^2$) | 398,199 (16.8%) | 1,887 (18.2%) | 73 (21.2%) | 688 (18.3%) | 1,126 (17.9%) |
| Obese (≥30 kg/m$^2$) | 157,579 (6.7%) | 832 (8.0%) | 40 (11.6%) | 357 (9.5%) | 435 (6.9%) |
| Missing | 591,700 (25.0%) | 1,944 (18.7%) | 76 (22.1%) | 761 (20.2%) | 1,107 (17.6%) |
| **Gestational hypertensive conditions[b]** | 100,011 (4.2%) | 1,192 (11.5%) | 33 (9.6%) | 531 (14.1%) | 628 (10.0%) |
| **Gestational diabetes mellitus** | 21,112 (0.9%) | 278 (2.7%) | 9 (2.6%) | 134 (3.6%) | 135 (2.2%) |
| **Birth weight for gestational age** | | | | | |
| Appropriate for gestational age | 2,213,837 (93.6%) | 9,385 (90.4%) | 298 (86.6%) | 3,310 (88.1%) | 5,777 (92.1%) |
| Small for gestational age | 56,558 (2.4%) | 172 (1.7%) | 19 (5.5%) | 91 (2.4%) | 62 (1.0%) |

*(Continued)*

**Table 1.** (Continued)

| | No ICP | ICP | Gestational week of ICP diagnosis | | |
|---|---|---|---|---|---|
| | | | <28 weeks | 28–36 weeks | ≥37 weeks |
| Large for gestational age | 82,215 (3.5%) | 739 (7.1%) | 17 (4.9%) | 310 (8.2%) | 412 (6.6%) |
| Missing | 12,868 (0.5%) | 82 (0.8%) | 10 (2.9%) | 48 (1.3%) | 24 (0.4%) |
| **Mode of delivery** | | | | | |
| Vaginal non-instrumental | 1,886,258 (79.7%) | 7,606 (73.3%) | 212 (61.6%) | 2,609 (69.4%) | 4,785 (76.3%) |
| Vaginal instrumental | 162,784 (6.9%) | 795 (7.7%) | 21 (6.1%) | 233 (6.2%) | 541 (8.6%) |
| Cesarean section | 316,436 (13.4%) | 1,977 (19.1%) | 111 (32.3%) | 917 (24.4%) | 949 (15.1%) |
| **Induction of labor** | | | | | |
| Spontaneous onset | 1,617,945 (68.4%) | 4,366 (42.1%) | 172 (50.0%) | 1,588 (42.2%) | 2,606 (41.5%) |
| Induced labor | 183,592 (7.8%) | 4,503 (43.4%) | 80 (23.3%) | 1,450 (38.6%) | 2,973 (47.4%) |
| Cesarean section before labor onset | 129,464 (5.5%) | 831 (8.0%) | 68 (19.8%) | 454 (12.1%) | 309 (4.9%) |
| Missing | 434,477 (18.4%) | 678 (6.5%) | 24 (7.0%) | 267 (7.1%) | 387 (6.2%) |
| **Gestational age at birth** | | | | | |
| Preterm (<37 weeks) | 116,683 (4.9%) | 1,405 (13.5%) | 88 (25.6%) | 1,317 (35.0%) | N/A |
| Term (37-<42 weeks) | 2,076,269 (87.8%) | 8,742 (84.2%) | 249 (72.4%) | 2,406 (64.0%) | 6,087 (97.0%) |
| Post-term (≥42 weeks) | 169,912 (7.2%) | 222 (2.1%) | 7 (2.0%) | 33 (0.9%) | 182 (2.9%) |
| Missing | 2,614 (0.1%) | 9 (0.1%) | 0 | <5 | 6 (0.1%) |
| **Apgar score at 5-minutes** | | | | | |
| ≥7 | 2,318,119 (98.0%) | 10,148 (97.8%) | 329 (95.6%) | 3,652 (97.2%) | 6,167 (98.3%) |
| <7 | 24,529 (1.0%) | 153 (1.5%) | 11 (3.2%) | 69 (1.8%) | 73 (1.2%) |
| Missing | 22,830 (1.0%) | 77 (0.7%) | <5 | 38 (1.0%) | 35 (0.6%) |
| **Neonatal asphyxia-related comorbidities** | 102,910 (4.4%) | 737 (7.1%) | 53 (15.4%) | 404 (10.7%) | 280 (4.5%) |
| **Neonatal hypoglycemia** | 42,838 (1.8%) | 383 (3.7%) | 10 (2.9%) | 221 (5.9%) | 152 (2.4%) |
| **Neonatal jaundice** | 107,386 (4.5%) | 719 (6.9%) | 49 (14.2%) | 467 (12.4%) | 203 (3.2%) |

All results are presented as numbers and percentages (*n*, %), with the denominator being the column total (Total *N*).

[a]Available from 1992 onwards.

[b]Including gestational hypertension, pre-eclampsia (mild and severe), and eclampsia.

BMI, body mass index; ICP, intrahepatic cholestasis of pregnancy.

year, birth month, maternal age, highest parental education level, maternal birth country, birth order, and maternal psychiatric history) slightly attenuated these associations. Specifically, children exposed to ICP had a 7.23% (95% CI 6.73 to 7.73) absolute risk of being diagnosed with neurodevelopmental conditions while those unexposed to ICP had a 6.05% (95% CI 6.02 to 6.08) absolute risk of being diagnosed, after adjusting for putative confounders (Fig 2). That is, children exposed to ICP were 22% more likely to be diagnosed with a neurodevelopmental condition (adjusted OR 1.22, 95% CI 1.13 to 1.31, $P < 0.001$), after controlling for putative confounders, compared to children born to mothers without ICP. These associations were consistent after controlling for shared familial factors between maternal full cousins (adjusted OR 1.24, 95% CI 1.09 to 1.42, $P = 0.001$), but attenuated after further controlling for factors shared between full siblings (adjusted OR 1.04, 95% CI 0.88 to 1.24, $P = 0.632$).

The strength of the association between ICP and offspring neurodevelopmental conditions increased the earlier ICP was diagnosed (Fig 2 and Table 2). For example, children born to mothers with ICP diagnosed <28 weeks of gestation were more than 2 times more likely to be diagnosed with a neurodevelopmental condition (adjusted OR 2.38, 95% CI 1.71 to 3.30, $P < 0.001$; adjusted absolute risk 13.93%, 95% CI 10.09% to 17.77%), as compared to children born to mothers without ICP, while children born to mothers with ICP diagnosed ≥37 weeks

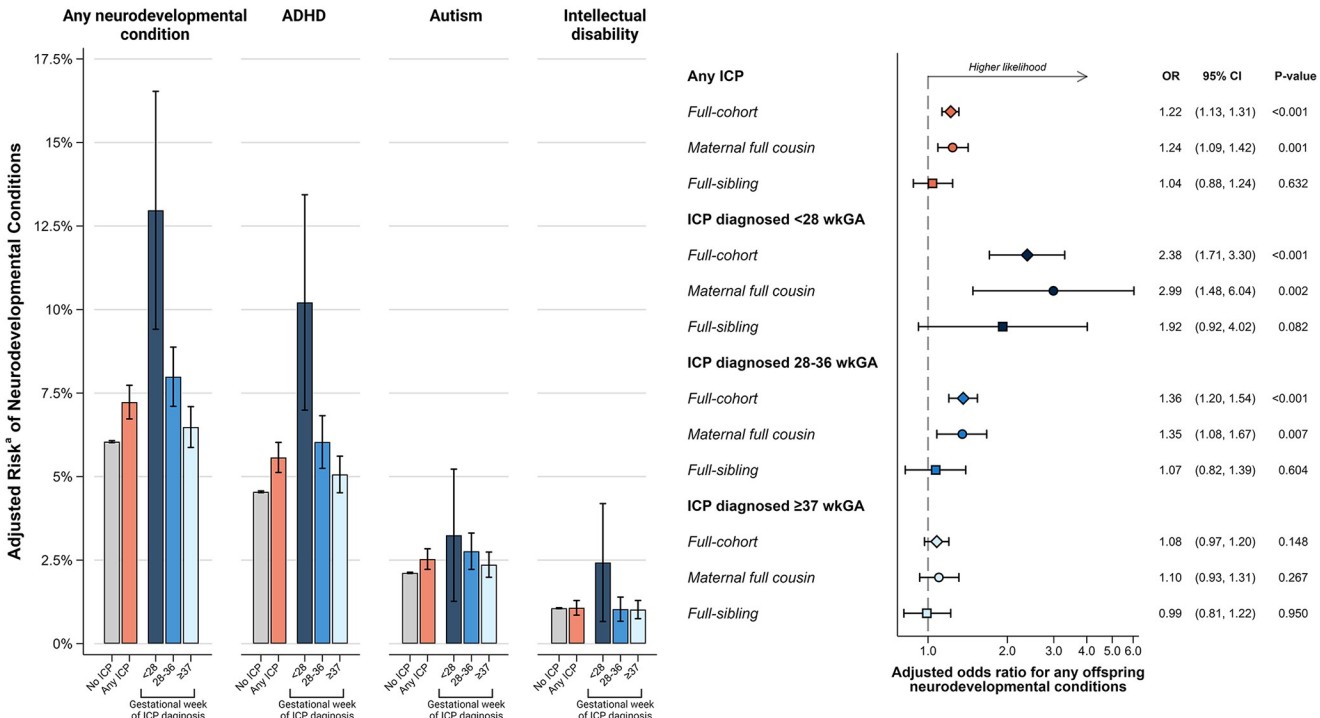

**Fig 2. The association (and 95% CI) between maternal intrahepatic cholestasis and offspring neurodevelopmental conditions, stratified by the gestational period of onset of intrahepatic cholestasis, with and without control for factors shared between maternal full cousins and full siblings.** All the models were adjusted for child's sex, birth year, maternal age, highest parental education level, maternal birth country, birth order, maternal psychiatric history, and birth month. For within-family analyses (i.e., maternal full-cousin analysis and full-sibling analysis), family-constant covariates were omitted. All standard errors were computed using the robust (sandwich) method. *P*-values were derived using Wald tests. [a]Marginal (counterfactual) probabilities calculated from logistic regressions. The error bars represent the 95% CIs in both panels. Figure created using BioRender.com, used under license/permission. ADHD, attention deficit/hyperactivity disorder; CI, confidence interval; ICP, intrahepatic cholestasis of pregnancy; wkGA, weeks of gestational age.

of gestation had comparable risk of neurodevelopmental conditions to that of children born to mothers without ICP (adjusted OR 1.08, 95% CI 0.97 to 1.20, *P* = 0.15; adjusted absolute risk 6.28%, 95% CI 5.68% to 6.88%). The association between early-onset ICP and offspring neurodevelopmental conditions was largely consistent after controlling for unobserved factors shared between full maternal cousins (adjusted OR 2.99, 95% CI 1.48 to 6.04, *P* = 0.002) and between full siblings (adjusted OR 1.92, 95% CI 0.92 to 4.02, *P* = 0.082), albeit with less precision.

For specific neurodevelopmental conditions, the association between ICP and offspring autism (adjusted OR 1.20, 95% CI 1.06 to 1.36, *P* = 0.004) and ADHD (adjusted OR 1.25, 95% CI 1.14 to 1.36, *P* < 0.001) were similar. However, the association between ICP and offspring intellectual disability was weaker and not statistically significant (adjusted OR 1.01, 95% CI 0.82 to 1.24, *P* = 0.91). When categorizing ICP based on the time of diagnosis in gestational weeks, the associations with ICP diagnosed before 28 weeks were similar for offspring ADHD (adjusted OR 2.46, 95% CI 1.70 to 3.55, *P* < 0.001) and intellectual disability (adjusted OR 2.34, 95% CI 1.10 to 4.96, *P* = 0.03), but the association with offspring autism was not statistically significant (adjusted OR 1.56, 95% CI 0.82 to 2.94, *P* = 0.17). For ICP diagnosed between 28 and 36 weeks, the associations with offspring autism (adjusted OR 1.32, 95% CI 1.07 to 1.62, *P* = 0.008) and ADHD (adjusted OR 1.36, 95% CI 1.18 to 1.57, *P* < 0.001) were similar, but there was not a statistically significant association with offspring intellectual disability (adjusted OR 0.97, 95% CI 0.68 to 1.39, *P* = 0.88). For ICP diagnosed at ≥37 weeks of gestation, none of

**Table 2. The association between maternal intrahepatic cholestasis and different offspring neurodevelopmental conditions, stratified by the gestational period of onset of intrahepatic cholestasis.**

| | N ICP Unexposed | N ICP Exposed | Unadjusted Odds ratio (95% CI) | Sex and birth year adjusted[a] Absolute risk unexposed % (95% CI) | Absolute risk exposed %, (95% CI) | Odds ratio (95% CI) | P-value | Fully adjusted model[a,b] Absolute risk unexposed % (95% CI) | Absolute risk exposed % (95% CI) | Odds ratio (95% CI) | P-value |
|---|---|---|---|---|---|---|---|---|---|---|---|
| **Any diagnoses of ICP** | | | | | | | | | | | |
| Total | 2,365,478 | 10,378 | | | | | | | | | |
| Any NDCs | 143,033 | 713 | 1.15 (1.06–1.24) | 6.05 (6.02–6.08) | 7.14 (6.64–7.64) | 1.20 (1.11–1.29) | <0.001 | 6.05 (6.02–6.08) | 7.23 (6.73–7.73) | 1.22 (1.13–1.31) | <0.001 |
| ADHD | 105,841 | 540 | 1.17 (1.08–1.28) | 4.55 (4.52–4.57) | 5.46 (5.01–5.91) | 1.22 (1.11–1.33) | <0.001 | 4.54 (4.52–4.57) | 5.57 (5.12–6.02) | 1.25 (1.14–1.36) | <0.001 |
| Autism | 48,111 | 252 | 1.20 (1.06–1.37) | 2.12 (2.10–2.14) | 2.58 (2.26–2.89) | 1.22 (1.08–1.39) | 0.002 | 2.12 (2.10–2.14) | 2.53 (2.22–2.83) | 1.20 (1.06–1.36) | 0.004 |
| Intellectual disability | 23,842 | 91 | 0.88 (0.71–1.08) | 1.06 (1.05–1.07) | 1.02 (0.81–1.23) | 0.96 (0.78–1.18) | 0.70 | 1.06 (1.05–1.07) | 1.07 (0.85–1.29) | 1.01 (0.82–1.24) | 0.91 |
| **ICP diagnosed <28 weeks** | | | | | | | | | | | |
| Total | 2,365,478 | 344 | | | | | | | | | |
| Any NDCs | 143,033 | 43 | 2.22 (1.61–3.06) | 6.05 (6.02–6.08) | 13.93 (10.09–17.77) | 2.55 (1.84–3.55) | <0.001 | 6.05 (6.02–6.08) | 12.97 (9.41–16.53) | 2.38 (1.71–3.30) | <0.001 |
| ADHD | 105,841 | 33 | 2.30 (1.61–3.30) | 4.55 (4.52–4.57) | 10.96 (7.46–14.46) | 2.62 (1.82–3.79) | <0.001 | 4.54 (4.52–4.57) | 10.21 (6.99–13.44) | 2.46 (1.70–3.55) | <0.001 |
| Autism | 48,111 | 10 | 1.54 (0.82–2.88) | 2.12 (2.10–2.14) | 3.52 (1.38–5.65) | 1.69 (0.90–3.19) | 0.11 | 2.12 (2.10–2.14) | 3.24 (1.27–5.22) | 1.56 (0.82–2.94) | 0.17 |
| Intellectual disability | 23,842 | 7 | 2.17 (1.03–4.59) | 1.06 (1.05–1.07) | 2.68 (0.73–4.63) | 2.57 (1.22–5.45) | 0.01 | 1.06 (1.05–1.07) | 2.43 (0.66–4.19) | 2.34 (1.10–4.96) | 0.03 |
| **ICP diagnosed between 28–36 weeks** | | | | | | | | | | | |
| Total | 2,365,478 | 3,759 | | | | | | | | | |
| Any NDCs | 143,033 | 277 | 1.24 (1.09–1.40) | 6.05 (6.02–6.08) | 8.08 (7.18–8.98) | 1.37 (1.21–1.55) | <0.001 | 6.05 (6.02–6.08) | 7.99 (7.10–8.88) | 1.36 (1.20–1.54) | <0.001 |
| ADHD | 105,841 | 204 | 1.23 (1.07–1.42) | 4.55 (4.52–4.57) | 6.07 (5.27–6.87) | 1.36 (1.18–1.57) | <0.001 | 4.54 (4.52–4.57) | 6.03 (5.25–6.82) | 1.36 (1.18–1.57) | <0.001 |
| Autism | 48,111 | 96 | 1.27 (1.04–1.56) | 2.12 (2.10–2.14) | 2.87 (2.30–3.43) | 1.37 (1.11–1.67) | 0.003 | 2.12 (2.10–2.14) | 2.77 (2.22–3.31) | 1.32 (1.07–1.62) | 0.008 |
| Intellectual disability | 23,842 | 31 | 0.83 (0.58–1.18) | 1.06 (1.05–1.07) | 1.02 (0.66–1.37) | 0.96 (0.67–1.37) | 0.82 | 1.06 (1.05–1.07) | 1.03 (0.67–1.39) | 0.97 (0.68–1.39) | 0.88 |
| **ICP diagnosed ≥37 weeks** | | | | | | | | | | | |
| Total | 2,365,478 | 6,275 | | | | | | | | | |
| Any NDCs | 143,033 | 393 | 1.04 (0.94–1.15) | 6.05 (6.02–6.08) | 6.28 (5.68–6.88) | 1.04 (0.94–1.16) | 0.43 | 6.05 (6.02–6.08) | 6.48 (5.87–7.09) | 1.08 (0.97–1.20) | 0.15 |
| ADHD | 105,841 | 303 | 1.08 (0.96–1.21) | 4.55 (4.52–4.57) | 4.86 (4.33–5.39) | 1.07 (0.96–1.21) | 0.23 | 4.54 (4.52–4.57) | 5.06 (4.52–5.61) | 1.12 (1.00–1.26) | 0.05 |

*(Continued)*

**Table 2.** (*Continued*)

| | N ICP Unexposed | N ICP Exposed | Unadjusted Odds ratio (95% CI) | Sex and birth year adjusted[a] | | | | Fully adjusted model[a,b] | | | |
|---|---|---|---|---|---|---|---|---|---|---|---|
| | | | | Absolute risk unexposed % (95% CI) | Absolute risk exposed %, (95% CI) | Odds ratio (95% CI) | *P*-value | Absolute risk unexposed % (95% CI) | Absolute risk exposed % (95% CI) | Odds ratio (95% CI) | *P*-value |
| *Autism* | 48,111 | 146 | 1.15 (0.97–1.35) | 2.12 (2.10–2.14) | 2.38 (2.00–2.76) | 1.13 (0.95–1.33) | 0.16 | 2.12 (2.10–2.14) | 2.36 (1.99–2.74) | 1.12 (0.95–1.32) | 0.18 |
| *Intellectual disability* | 23,842 | 53 | 0.84 (0.64–1.10) | 1.06 (1.05–1.07) | 0.94 (0.69–1.20) | 0.89 (0.68–1.16) | 0.39 | 1.06 (1.05–1.07) | 1.02 (0.75–1.29) | 0.96 (0.73–1.26) | 0.78 |

[a]Logistic regression models with robust (sandwich) standard errors. *P*-values were derived using Wald tests.

[b]Adjusted for sex, birth year, maternal age, highest parental education level, maternal birth country, birth order, maternal psychiatric history, and birth month.

ADHD, attention deficit/hyperactivity disorder; ICP, intrahepatic cholestasis of pregnancy; NDC, neurodevelopmental conditions.

the associations with specific neurodevelopmental outcomes were statistically significant (ADHD adjusted OR 1.12, 95% CI 1.00 to 1.26, *P* = 0.05; autism adjusted OR 1.12, 95% CI 0.95 to 1.32, *P* = 0.18; intellectual disability adjusted OR 0.96, 95% CI 0.73 to 1.26, *P* = 0.78) (Fig 2 and Table 2).

Adjusting for gestational age slightly attenuated the observed associations but did not alter the qualitative conclusion; early ICP was associated with neurodevelopmental conditions irrespective of controlling for gestational age (Table 3), albeit with slightly less precision. For example, further adjusting for gestational age altered the statistical significance of the association between ICP diagnosed before 28 weeks and intellectual disability, as well as the association between ICP diagnosed between 28 and 36 weeks and autism (Table 3).

## Sensitivity analyses

Results from sensitivity analyses were consistent with those from primary analyses (S3–S4 Tables and S1–S4 Figs). That is, the associations remained consistent with the main findings after excluding individuals without identifiable maternal cousins (*N* = 1,584,097) or full siblings (*N* = 664,358) (*P*-values of cross-model Wald tests ≥ 0.05) (S3 Table). The associations were also consistent with the main findings when ICP cases diagnosed on the delivery date (*N* = 2,290) were excluded (*P*-values of cross-model Wald tests ≥ 0.05) (S4 Table). However, after excluding preterm births (*N* = 118,088), the association between ICP diagnosed before 28 weeks of gestation and offspring intellectual disability was no longer statistically significant, though other estimates remained largely in line with the main findings (S4 Table). The results were slightly attenuated after excluding individuals born before 1992 (*N* = 553,230). While the majority of the associations were still statistically significant, the association between ICP diagnosed before 28 weeks of gestation and offspring intellectual disability was no longer statistically significant (adjusted OR 1.85, 95% CI 0.76 to 4.52) (S5 Table). After further adjustment for maternal BMI (irrespective of via complete case analysis, "missing-as-indicator," multiple imputation or inverse-probability weights), slight attenuations were observed in the associations between maternal ICP diagnosed before 28 weeks and between 28 and 36 weeks of gestation, and offspring neurodevelopmental conditions. Similarly, the association between ICP diagnosed before 28 weeks of gestation and offspring intellectual disability was not statistically significant (adjusted OR$_{\text{complete case analysis}}$ 1.88, 95% CI 0.69 to 5.08; adjusted OR$_{\text{missing-as-indicator}}$ 1.81, 95% CI 0.74 to 4.41; adjusted OR$_{\text{inverse-probability weights}}$ 1.84, 95% CI 0.51 to 6.66; adjusted OR$_{\text{multiple imputation}}$ 1.82, 95% CI 0.75 to 4.44) (S5 Table).

**Table 3. The association between intrahepatic cholestasis of pregnancy and neurodevelopmental conditions after adjusting for gestational age.**

| | Adjusting for gestational age as: | | | | | |
|---|---|---|---|---|---|---|
| | Linear term[a] | | Categorical term[b] | | Restricted cubic splines[c] | |
| | OR (95% CI) | P-value | OR (95% CI) | P-value | OR (95% CI) | P-value |
| **Any diagnoses of ICP** | | | | | | |
| Any NDCs | 1.15 (1.06–1.24) | 0.001 | 1.14 (1.05–1.23) | 0.001 | 1.15 (1.07–1.24) | <0.001 |
| ADHD | 1.19 (1.09–1.30) | <0.001 | 1.18 (1.08–1.29) | <0.001 | 1.19 (1.09–1.30) | <0.001 |
| Autism | 1.14 (1.00–1.29) | 0.045 | 1.13 (1.00–1.28) | 0.056 | 1.14 (1.01–1.30) | 0.037 |
| Intellectual disability | 0.88 (0.71–1.08) | 0.228 | 0.84 (0.68–1.03) | 0.099 | 0.89 (0.72–1.09) | 0.255 |
| **Diagnosed <28 weeks** | | | | | | |
| Any NDCs | 2.13 (1.53–2.97) | <0.001 | 2.14 (1.54–2.98) | <0.001 | 2.09 (1.50–2.91) | <0.001 |
| ADHD | 2.27 (1.57–3.28) | <0.001 | 2.28 (1.58–3.30) | <0.001 | 2.24 (1.55–3.24) | <0.001 |
| Autism | 1.39 (0.73–2.64) | 0.311 | 1.40 (0.74–2.66) | 0.297 | 1.36 (0.71–2.58) | 0.353 |
| Intellectual disability | 1.73 (0.81–3.70) | 0.157 | 1.75 (0.82–3.71) | 0.145 | 1.68 (0.78–3.60) | 0.183 |
| **Diagnosed between 28–36 weeks** | | | | | | |
| Any NDCs | 1.20 (1.06–1.36) | 0.004 | 1.14 (1.01–1.30) | 0.034 | 1.19 (1.05–1.34) | 0.007 |
| ADHD | 1.24 (1.07–1.43) | 0.004 | 1.20 (1.04–1.38) | 0.014 | 1.22 (1.06–1.41) | 0.006 |
| Autism | 1.17 (0.96–1.44) | 0.125 | 1.12 (0.92–1.38) | 0.262 | 1.16 (0.94–1.42) | 0.159 |
| Intellectual disability | 0.72 (0.51–1.03) | 0.076 | 0.62 (0.43–0.88) | 0.008 | 0.71 (0.50–1.02) | 0.062 |
| **Diagnosed ≥37 weeks** | | | | | | |
| Any NDCs | 1.06 (0.96–1.18) | 0.265 | 1.07 (0.97–1.19) | 0.171 | 1.08 (0.97–1.19) | 0.161 |
| ADHD | 1.11 (0.99–1.24) | 0.086 | 1.11 (0.99–1.25) | 0.075 | 1.12 (0.99–1.25) | 0.067 |
| Autism | 1.10 (0.93–1.30) | 0.248 | 1.12 (0.95–1.32) | 0.178 | 1.12 (0.95–1.32) | 0.170 |
| Intellectual disability | 0.94 (0.71–1.23) | 0.637 | 0.97 (0.74–1.28) | 0.849 | 0.96 (0.73–1.26) | 0.785 |

[a]Logistic regression models with standard errors computed using the robust (sandwich) method. Adjusted for the same covariates as the main model (i.e., child's sex, birth year, maternal age, highest parental education level, maternal birth country, birth order, maternal psychiatric history, and birth month) and gestational age in weeks as a linear term.

[b]Adjusted for the same covariates as the main model and gestational age in weeks as a categorical term (<37, 37-, 38-, 39-, 40-, 41-, 42-).

[c]Adjusted for the same covariates as the main model and gestational age in weeks using restricted cubic splines with 4 knots placed at the 5th, 35th, 65th, and 95th percentile.

ADHD, attention deficit/hyperactivity disorder; ICP, intrahepatic cholestasis of pregnancy; NDC, neurodevelopmental disorder.

Similar to findings relating to early-onset ICP, a longer duration of exposure to maternal ICP was associated with greater odds of offspring neurodevelopmental conditions (S1 Fig). That is, children exposed to ICP for more than 16 weeks were at increased odds of being diagnosed with neurodevelopmental conditions (OR 2.90, 95% CI 1.45 to 5.80), as compared to the children who were never exposed to ICP. However, all children in this group (duration >16 weeks) were diagnosed with ICP before 28 weeks of gestation. Modeling the time of diagnosis of ICP as continuous variables with restricted cubic splines (S2 Fig), or as the completed percentage of pregnancy at ICP diagnosis (S3 Fig), among women with ICP yielded results in line with our main finding that earlier diagnosis of ICP was more strongly associated with offspring neurodevelopment. A similar pattern was also observed in the sibling analysis using continuous variables (S2 Fig). Restricting the analysis to those with ICP and contrasting the associations between categories of onset of ICP yielded the same conclusion as our main analysis (S4 Fig).

## Discussion

In this Swedish nationwide cohort study, we found that children born to mothers with ICP were more likely to be diagnosed with neurodevelopmental conditions than children born to

mothers without ICP. Specifically, children born to mothers diagnosed with ICP before 28 weeks of gestation were twice as likely to be diagnosed with neurodevelopmental conditions. Although we found some evidence to suggest that genetic and environmental factors shared between full siblings and full maternal cousins contribute to this relationship, these factors did not appear to fully explain the higher occurrence of neurodevelopmental conditions among children exposed to early-onset ICP.

Though it is well recognized that maternal ICP is a risk factor for obstetric and neonatal complications [4,19] and that the risk increases with the concentration of serum bile acid levels [24], we are unaware of any previous studies that have studied the long-term effects of maternal ICP on offspring neurodevelopmental conditions. Potential mechanistic pathways linking maternal ICP and offspring neurodevelopmental conditions include placental vasospasms and fetal hypoxia [25–27], oxidative stress [28–30], and chronic inflammation [31–34]. Persons with early-onset ICP had a higher proportion of SGA, cesarean section, low Apgar score at 5-minutes, neonatal asphyxia-related comorbidities, and neonatal jaundice. These factors are associated with increased risks of neurodevelopmental conditions in offspring [16–18,20]. Another potential mediator linking ICP to offspring neurodevelopment is preterm birth. However, we did not observe a higher proportion of preterm births among those diagnosed with early-onset ICP. Furthermore, excluding premature births or adjusting for gestational age at birth only had a negligible effect on most of our estimates, suggesting that preterm birth is unlikely to explain the stronger associations observed with early-onset ICP. However, adjusting for gestational age altered the statistical significance of the association between ICP diagnosed before 28 weeks and intellectual disability, as well as the association between ICP diagnosed between 28 and 36 weeks and autism. It is important to note, however, that adjusting for gestational age might introduce bias into the results, as it can act as either a mediator or a collider (refer to S5 Fig). Therefore, caution is necessary when interpreting results that have been adjusted for gestational age. It is important, however, to emphasize that preterm birth is a well-recognized risk factor for offspring neurodevelopmental conditions [16–18]. Our findings regarding the length of exposure to ICP should not be misconstrued as an encouragement for earlier deliveries of these children. In fact, although we are not powered to formally examine mediation, it seems probable that the connection between early ICP and offspring neurodevelopmental conditions may involve shorter gestation; earlier (iatrogenic) delivery is not necessarily recommended because it may partly explain the observed excess risk, and we currently do not know if treatment (i.e., ursodeoxycholic acid) might mitigate the observed risks. Further investigations are warranted to better understand the underlying etiology of early-onset ICP and the possible mechanisms that implicate it in offspring neurodevelopmental conditions.

The primary strength of this study is our large nationwide sample, which allowed us to examine a population with minimal selection bias and loss to follow-up and to utilize robust methods to mitigate unobserved genetic and environmental confounders—strengthening our causal inferences beyond that of traditional methods.

There are, however, some limitations of using Swedish health registries that should be acknowledged. First, although a longer duration of exposure (duration ≥17 weeks) to ICP was associated with higher risks of neurodevelopmental conditions, all cases with ICP exposure ≥17 weeks were diagnosed with ICP before 28 weeks of gestation. Therefore, we were not able to clearly differentiate whether this association was due to ICP exposure during a critical window in early fetal development or the consequence of greater cumulative duration of fetal exposure to ICP. Nonetheless, these analyses suggest that ICP is somehow involved in the etiology of neurodevelopmental conditions in the offspring. Second, the registers do not record any measures of bile acid concentration, so we were unable to examine the role of serum bile

acid concentration—which is a strong marker for ICP severity. Third, ursodeoxycholic acid had not gained widespread use as a treatment for ICP in Sweden during our study period, and we were therefore unable to examine its potential role. We suggest that future studies examine if ursodeoxycholic acid may mitigate the role of ICP in offspring neurodevelopment. Fourth, although we were able to control for a range of both observed and unobserved confounders, our study is observational and may be subject to residual confounding. Fifth, while our study encompasses almost everyone giving birth in Sweden, it is not certain that our findings generalize to other populations with different prevalence's of ICP and neurodevelopmental conditions. Finally, although our findings point to a role of early-onset ICP in neurodevelopment, ICP rarely presents in early gestation, with only a limited number of children exposed to ICP before 28 weeks of gestation in this study. While we have made thorough efforts to describe women with early-onset ICP and their obstetric and neonatal complications, which we attempt to control for, they may not be comparable to other women who develop ICP—differing in characteristics above and beyond what can be controlled for in register-based studies. To resolve this, we urge replication of our study, including collection of more detailed information on women experiencing early-onset ICP in future studies.

Of note, early-onset ICP only accounted for 3.3% of the ICP cases in our study sample as most ICP cases were diagnosed in the late second or third trimester [1,19]. Clinically, this means that few women and children are likely to ever experience this exposure (0.16% of the population) and that eliminating this exposure from the population would only have a small impact on the population-wide prevalence of neurodevelopmental conditions.

In this large population-based cohort study, we found evidence to suggest that offspring exposed to ICP during pregnancy are more likely to be diagnosed with neurodevelopmental conditions, especially when exposed to early-onset ICP. Further studies are warranted to replicate our findings and to explore the potential underlying mechanisms of this relationship.

## Supporting information

**S1 Appendix. STROBE Checklist.** Strengthening the Reporting of Observational Studies in Epidemiology (STROBE) checklist.
(DOCX)

**S2 Appendix. Protocol.** Prospectively recorded analysis plan.
(DOCX)

**S1 Fig. The association between duration of fetal exposure to maternal intrahepatic cholestasis and any offspring neurodevelopmental conditions, and the distribution of week of onset across duration of exposure.**
(DOCX)

**S2 Fig. The association between gestational week of maternal intrahepatic cholestasis diagnosis and any offspring neurodevelopmental conditions, among the offspring exposed to maternal intrahepatic cholestasis ($N$ = 10,378), separated by full-cohort analysis and full-sibling analysis.**
(DOCX)

**S3 Fig. The association between timing at maternal intrahepatic cholestasis diagnosis, as a function of the percentage of pregnancy completed at diagnosis, and any offspring neurodevelopmental conditions among children exposed to ICP ($N$ = 10,378).**
(DOCX)

**S4 Fig. The association between the categorical timing of intrahepatic cholestasis of pregnancy diagnosis and any neurodevelopmental conditions in offspring among those who were born to mothers with ICP (*N* = 10,378).**
(DOCX)

**S5 Fig. Directed Acyclic Graph illustrating the adjustments for gestational age.**
(DOCX)

**S1 Table. Extended details on variable definitions and their underlying ICD/ATC codes.**
(DOCX)

**S2 Table. Characteristics of the study sample over diagnoses of neurodevelopmental conditions in offspring.**
(DOCX)

**S3 Table. The association between intrahepatic cholestasis of pregnancy and any neurodevelopmental conditions among those with identifiable full cousins and full siblings.**
(DOCX)

**S4 Table. The association between intrahepatic cholestasis of pregnancy and neurodevelopmental conditions after excluding those with intrahepatic cholestasis of pregnancy diagnosed at delivery and after excluding those born prematurely.**
(DOCX)

**S5 Table. The association between intrahepatic cholestasis of pregnancy and neurodevelopmental conditions. The association between intrahepatic cholestasis of pregnancy and neurodevelopmental conditions while adjusting for maternal BMI.**
(DOCX)

## Author Contributions

**Conceptualization:** Shuyun Chen, Viktor H. Ahlqvist, Brian K. Lee, Renee M. Gardner.

**Data curation:** Hugo Sjöqvist.

**Formal analysis:** Shuyun Chen, Viktor H. Ahlqvist.

**Funding acquisition:** Christina Dalman, Renee M. Gardner.

**Methodology:** Shuyun Chen, Viktor H. Ahlqvist, Hugo Sjöqvist, Olof Stephansson, Cecilia Magnusson, Christina Dalman, Håkan Karlsson, Brian K. Lee, Renee M. Gardner.

**Resources:** Christina Dalman.

**Supervision:** Brian K. Lee, Renee M. Gardner.

**Visualization:** Shuyun Chen, Viktor H. Ahlqvist.

**Writing – original draft:** Shuyun Chen, Viktor H. Ahlqvist.

**Writing – review & editing:** Shuyun Chen, Viktor H. Ahlqvist, Hugo Sjöqvist, Olof Stephansson, Cecilia Magnusson, Christina Dalman, Håkan Karlsson, Brian K. Lee, Renee M. Gardner.

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
