## [Editor Report · Decision Letter 0]

12 Jul 2023

Dear Dr Ahlqvist, 

Thank you for submitting your manuscript entitled "Maternal intrahepatic cholestasis of pregnancy and neurodevelopmental conditions in offspring: a population-based cohort study of two million Swedish children" for consideration by PLOS Medicine.

Your manuscript has now been evaluated by the PLOS Medicine editorial staff with relevant expertise and I am writing to let you know that we would like to send your submission out for external peer review. Please include continuous line numbers in your revised manuscript (i.e. not starting from 1 with each new page).

Please re-submit your manuscript within two working days, i.e. by Jul 14 2023 11:59PM.

Kind regards,

Louise Gaynor-Brook, MBBS PhD

Senior Editor

PLOS Medicine

---

## [Decision Letter · Decision Letter 1]

28 Sep 2023

Dear Dr. Ahlqvist,

Thank you very much for submitting your manuscript "Maternal intrahepatic cholestasis of pregnancy and neurodevelopmental conditions in offspring: a population-based cohort study of two million Swedish children" (PMEDICINE-D-23-01940R1) for consideration at PLOS Medicine. 

Your paper was evaluated by four independent reviewers, including a statistical reviewer, and was discussed among all the editors here and with an academic editor with relevant expertise. The reviews are appended at the bottom of this email and any accompanying reviewer attachments can be seen via the link below:

[LINK]

In light of these reviews, I am afraid that we will not be able to accept the manuscript for publication in the journal in its current form, but we would like to consider a revised version that addresses the reviewers' and editors' comments. Obviously we cannot make any decision about publication until we have seen the revised manuscript and your response, and we plan to seek re-review by one or more of the reviewers. 

We expect to receive your revised manuscript by Oct 19 2023 11:59PM. Please email me (lgaynor@plos.org) if you have any questions or concerns.

We look forward to receiving your revised manuscript. 

Sincerely,

Louise Gaynor-Brook, MBBS PhD

lgaynor@plos.org

plosmedicine.org

Thank you for your patience, and apologies for the delay in providing you with an editorial decision. As you will see from the reviewer comments, adjustment is required for gestational age in your analyses. Please respond to the reviewer and editorial comments in full before resubmitting your manuscript, which will be re-reviewed. 

Comment from the Academic Editor:

The authors need to fully adjust for GA; please see e.g. PMID 20543995 which addresses the continuous relationship between GA and special educational needs.

General comments:

Throughout the paper, please adapt reference call-outs to the following style: "... every year [1,2]." (noting the absence of spaces within the square brackets).

Please define all abbreviations at first use e.g. CI, IQR, etc. 

Abstract:

Please structure your abstract using the PLOS Medicine headings (Background, Methods and Findings (combined in one subsection), Conclusions). Please ensure that the Abstract is written as prose. 

Abstract Background: The final sentence should clearly state the study question.

Abstract Methods and Findings:

Please provide brief demographic details of the study population (e.g. sex, ethnicity, etc)

Please include the important dependent variables that are adjusted for in the analyses.

In the last sentence of the Abstract Methods and Findings section, please describe 2-3 of the main limitations of the study's methodology.

Please define CI at first use.

Please define “term” in terms of gestational age. 

Please clarify what is being compared for the ORs presented for maternal cousins and full siblings. 

Abstract Conclusions:

Please begin your Abstract Conclusions with "In this study, we observed ..." or similar, to summarize the main findings from your study, without overstating your conclusions. Please emphasize what is new and address the implications of your study, being careful to avoid assertions of primacy. 

Author Summary:

In the final bullet point of ‘What Do These Findings Mean?’, please describe the main limitations of the study in non-technical language.

Introduction:

Line 77 - please temper assertions of primacy by adding ‘to the best of our knowledge’ or similar 

Methods:

Did your study have a prospective protocol or analysis plan? Please state this (either way) early in the Methods section. If a prospective analysis plan (from your funding proposal, IRB or other ethics committee submission, study protocol, or other planning document written before analyzing the data) was used in designing the study, please include the relevant prospectively written document with your revised manuscript as a Supporting Information file to be published alongside your study, and cite it in the Methods section. A legend for this file should be included at the end of your manuscript. If no such document exists, please make sure that the Methods section transparently describes when analyses were planned, and if/when reported analyses differed from those that were planned. Changes in the analysis-- including those made in response to peer review comments-- should be identified as such in the Methods section of the paper, with rationale. If a reported analysis was performed based on an interesting but unanticipated pattern in the data, please be clear that the analysis was data-driven.

Please add the following statement, or similar, to the Methods: "This study is reported as per the Strengthening the Reporting of Observational Studies in Epidemiology (STROBE) guideline (S1 Checklist)." The STROBE guideline can be found here: http://www.equator-network.org/reporting-guidelines/strobe/

Results: 

Please present numerators and denominators for percentages in the Tables.

For the adjusted analyses in Table 2, please also provide the unadjusted analyses.

Please indicate which factors are adjusted for, in the main text.

Line 209 - please revise to ‘strength of the association’

Line 221 - the results presented in this paragraph do not correspond well with the results presented in Table 2, showing that there is only an association (after full adjustment) between ICP and autism between 28-36 weeks of pregnancy (and all diagnoses of ICP); and between ICP before 28 weeks of pregnancy and intellectual disability . Please revise this paragraph to better reflect the results presented in Table 2, especially the sentence “children born to mothers with ICP diagnosed before 28 weeks of gestation were more likely to be diagnosed with.. autism (adjusted OR 1.56, 95% CI 0.82-2.94)"

Please see https://journals.plos.org/plosmedicine/s/supporting-information for our supporting information 

guidelines; supplementary files should be referred to as Table S1, S2, Figure S1, S2, and so on. 

Line 230 - please provide the results for the analysis excluding those without siblings (N=515 550) in eTable4

Sensitivity analyses: Please revise this paragraph to reflect that associations were not similar for ICP before 28 weeks of pregnancy and intellectual disability when excluding preterm birth, children born before 1993 and after adjustment for maternal BMI. 

Discussion:

Please present and organize the Discussion as follows: a short, clear summary of the article's findings; what the study adds to existing research and where and why the results may differ from previous research; strengths and limitations of the study; implications and next steps for research, clinical practice, and/or public policy; one-paragraph conclusion.

Please remove all subheadings within your Discussion e.g. Strengths and limitations 

Figures:

When a p value is given, please specify the statistical test used to determine it in the figure legend. 

Please define abbreviations used in the figure legend of each figure.

Please indicate in the figure caption what is represented by the error bars. 

Tables:

When a p value is given, please specify the statistical test used to determine it in the table legend.

Please define all abbreviations used in the table legend of each table.

Please provide the unadjusted comparisons as well as the adjusted comparisons in Table 2. 

References:

Please ensure that journal name abbreviations match those found in the National Center for Biotechnology Information (NCBI) databases (http://www.ncbi.nlm.nih.gov/nlmcatalog/journals), and are appropriately formatted and capitalised. Six authors should be listed prior to ‘et al’. 

Please also see https://journals.plos.org/plosmedicine/s/submission-guidelines#loc-references for further details on reference formatting. 

Comments from the reviewers:

Reviewer #1: Review of manuscript:

"Maternal intrahepatic cholestasis of pregnancy and neurodevelopmental conditions in offspring: a population-based cohort study of two million Swedish children"

Thank you for inviting me to review this manuscript, which uses a retrospective national dataset and linked coded diagnostic terms to interrogate any associations between ICP and offspring neurodevelopmental disorders (NDD). It is a useful and important topic, which attempts to answer a question that has not previously been adequately addressed within the literature, and also addresses questions that I have heard asked by affected families within PPI engagement. 

This study makes use of the excellent Swedish data collection at a population level, and is an appropriate population and study design to attempt to address this question. 

It is important to recognise that, once this study is published, its simplest message will be interpreted across the patient community as being that "ICP causes NDD", which still cannot be proven by this retrospective approach demonstrating association rather than causation, and thus there is a critical importance that all co-variates are accounted for appropriately, and a very clear message is presented through from the abstract about this limitation.

The authors have taken multiple approaches to account for the impact of co-variates on their overall findings. However, the major concern that I have with their findings is that they have not appropriately accounted for gestational age at birth - excluding those babies born preterm simply does not exclude the impact of advancing gestational age at birth being inversely associated with NDD even after 37/40. There is a plethora of studies demonstrating this, and it is one of the main counselling factors underpinning recommendations regarding planned gestational age at birth; Swedish population studies have also beautifully demonstrated the shorter term impacts of early term birth on many neonatal outcomes that plausibly could affect later NDD (Mitha et al. J Pediatrics 2021; 223: 43-50.e5). 

I therefore would recommend that the authors include gestational age at birth as a continuous variable in their statistical models to confirm that this does not impact their conclusion before considering accepting the manuscript. I have huge concerns that the risk of publishing this manuscript without absolute certainty of the findings will be that patients read the association between longer exposure to ICP and increased risk of NDD, and therefore request earlier birth to prevent this (which in actual fact might increase the risk of NDD and other consequences of preterm and early term iatrogenic birth). We know that patients with early onset ICP are more likely to give birth at earlier gestations, and so this finding is not unsurprising. 

Similarly, it is very easy to look through the (well presented) Table 1 and Supplementary Table and come to vastly different conclusions if all of these factors are not accounted for in the final analyses: I would suggest from these data that:

1) More recent birth year

2) Caesarean birth

3) Induction of labour

4) Overweight / obesity or missing BMI data

Etc etc may be more associated with ICP and / or NDD - the authors have accounted for many of these factors in these models but it would be worth summarising in a table / figure which factors are independently associated with autism in separate models so that you are able to clearly follow why you have selected your various sensitivity analyses and confounders included. Otherwise, it may appear by chance that you have selected BMI as the factor to adjust by (supp table 4) rather than through logical selection of the most important confounding variable. 

In summary, I think that this is an important and useful manuscript, but would not recommend publication in its current format without appropriate correction as described above, most importantly treating gestation of birth as a continuous variable. Once published, the manuscript risks reactionary concerns from those patients and clinicians who will read only the abstract and headline conclusion, and risk detrimental care decisions for the pregnancy which may be difficult to reverse given the relatively limited datasets with this duration of follow up that currently could be used to address this question.

Reviewer #2: Thanks for the opportunity to review your manuscript. My role is as a statistical reviewer, so my review concentrates on the study design, data, and analysis that are presented. I have put general questions first, followed by queries relevant to a specific section of the manuscript (with a page/paragraph reference).

This manuscript examines whether exposure to intrahepatic cholestasis during gestation is associated with later incidence of neurodevelopmental conditions (ADHD, autism, ID). The link between is ICP and ND is proposed based on links between ICP and other neonatal adverse events, and assuming ICP could similarly alter fetal brain development during a critical period. Data is taken from linked routinely collected healthcare data and registries from Sweden, and initially includes all children born between 1987 and 2010. Children not in the birth registry, or were not in country (and with data) for less than 5 years, non-singletons were excluded. Children with missing covariate information were also excluded (0.1 % of initial sample). ICP (exposure) was identified from inpatient data, with date of diagnosis used for date of onset (and categorised into three categories). ND conditions (outcome) were ascertained from inpatient data, and dispensing for drugs used to treat ADHD. A combined ND outcome was used, as well as specific diagnoses as secondary outcomes. A range of covariates is considered, these are sourced from several different databases and includes in maternal, gestational, and birth information. Full details (i.e. ICD-10 codes) are provided in the appendix. The main analysis uses logistic regression, one minimally adjusted with sex and year of birth, and one that also considers more potential confounders. Several sensitivity analyses of the main analysis are considered, the rationale for these is explained well. In several secondary analyses, a conditional LR analysis of siblings and cousins who were discordant for ICP status were completed. 

There is a small association between ICP and ND in the main analysis, this shrinks back to no association in the full-sibling analysis. There is a 'dose-response' with period of ICP exposure, with more ND seen in those exposed to ICP from earlier in development. The association between ND and ICP seemed to be mostly driven by ADHD and Autism, with no robust association between ICP and ID. The results from the sensitivity analyses were consistent with the main analyses. Limitations of the study (e.g. limitations in the data collection) are clearly discussed. The figures presented in the manuscript are excellent - very clear and elegant. 

P3, L75. I'm not sure what the right wording here is, but stillbirth can't be an intermediary between ICP and ND conditions. '

P4, L94. Why wasn't more recent data (2011-) used in the study? 

P4, L96. Were the children identified but on the birth registry likely to have been born outside of Sweden? Or are these non-matches from the data linkage process? 

P5, L107. Would this condition typically be diagnosed soon after onset? e.g. is it symptomatic or part of normal screening? 

P5, L118. Is atomexetine used for paediatric cases of narcolepsy? i.e. would all dispensing of this drug be strictly for ADHD? 

P6, L137. Were robust standard errors used to account for multiple children born to the same mother? 

P6, L139. The second model uses a subset of the potential confounders detailed in the previous section. How were these particular covariates selected for the second model? 

P6, L165. Missing-as-indicator assumes no correlation between the exposure and the missing variable - a more robust option here would be to repeat the analysis with something like IPW or MI that is robust for data that is missing-at-random (i.e. conditional on the observed data). 

Figure 1. In D), what method was used to create the smoothed incidence curves over age? 

Reviewer #3: This is a nice paper. Original research. No suggestions for improvement. 

I recommend acceptance. 

Jim Thornton

Nottingham. 31 July 2023

Reviewer #4: I am asked to comment on revision 1 of this article.

My main concern is the correction for the main confounder gestational age. As ICP results more often in preterm birth (as the authors acknowledge) this should be controlled for.

The authors write

"and our main model, where we further adjust for birth month, birth order, maternal age, maternal country of birth and maternal psychiatric history and the highest parental education level."

Gestational age is not mentioned here.

Gestational age is reported as <37, 37-42 and >42 weeks.

They then provide data stratified for gestational age at diagnosis, by reporting them in any, and diagnosis <28, 28-37 and > 37. The stronger effect after the earlier diagnosis is to be expected as these pregnancies will end earlier.

I am afraid the correction for gestational age (if it already takes place?) in categories <37, 37-42 and >42 weeks is insufficient as within these categories the GA of the ICP group will be younger.

After these analyses I am happy to review again, but for me at the moment this difference in gestational age might be the main explanation/confounder for what we see here.

Prof Ben W Mol

[LINK]

---

## [Decision Letter · Decision Letter 2]

22 Nov 2023

Dear Dr. Ahlqvist,

Thank you very much for re-submitting your manuscript "Maternal intrahepatic cholestasis of pregnancy and neurodevelopmental conditions in offspring: a population-based cohort study of two million Swedish children" (PMEDICINE-D-23-01940R2) for review by PLOS Medicine.

I have discussed the paper with my colleagues and the academic editor and it was also seen again by three reviewers. I am pleased to say that provided the remaining editorial and production issues are dealt with we are planning to accept the paper for publication in the journal.

[LINK]

We expect to receive your revised manuscript within 1 week. Please email me(lgaynor@plos.org) if you have any questions or concerns.

If you have any questions in the meantime, please contact me on lgaynor@plos.org.  

We look forward to receiving the revised manuscript by Nov 29 2023 11:59PM.   

Sincerely,

Louise Gaynor-Brook, MBBS PhD

plosmedicine.org

Requests from Editors:

Thank you for your patience, and apologies for the delay in providing you with an editorial decision. The list below contains minor points which should not require a substantial amount of time to attend to. 

After discussion among the editorial team and with the Academic Editor, and in line with Reviewer 1, we request that S6 Table (adjusting for gestational age) is incorporated into the main paper. 

To help us extend the reach of your research, please provide any Twitter handle(s) that would be appropriate to tag, including your own, your co-authors', your institution, funder, etc.

Abstract:

Line 63 - In view of reviewer concerns, we suggest revising the first sentence of your Conclusions to “In this study, we observed that exposure to ICP during gestation is associated with an increased likelihood of neurodevelopmental conditions in offspring, particularly in cases of early-onset ICP” - or similar. 

Author Summary:

Line 106 - We suggest revising to “Diagnosis of ICP during pregnancy, especially early in pregnancy, is associated with an increased likelihood of neurodevelopmental disorders in the children exposed during gestation” or similar 

Line 109 - Please revise ‘causal relationship’. We suggest “...it is not possible to determine whether ICP is a causative factor in the development of neurodevelopmental conditions in children born to affected mothers” or similar 

Introduction:

Line 133 - please revise to ‘...by examining the association between…’

Line 139 - please revise to ‘differentially affects the likelihood of’

Methods:

Please refer to your prospective analysis plan early in the Methods section. 

Results: 

Line 265 - please revise to Children born to mothers with ICP

Please provide p values alongside 95% CIs where available.

Line 309 - please clarify whether this is adjusted OR

Line 354 - As you have been so transparent about where results are no longer statistically significant, it is appropriate to comment that adjusting for gestational age affects the statistical significance of the association between intellectual disability and ICP diagnosed <28w, and between autism and ICP diagnosed between 28-36 weeks / any diagnoses of ICP.

Line 356 - Please incorporate S6 Table into the main paper 

Discussion:

Line 379 - please revise that adjusting for gestational age had only a negligible effect on the estimates. It would be preferable to acknowledge that adjusting for gestational age affected the statistical significance of the association between intellectual disability and ICP diagnosed <28w, and between autism and ICP diagnosed between 28-36 weeks / any diagnoses of ICP; and to reiterate what is demonstrated in Figure S5.

Line 402 - please replace effect with association

Comments from Reviewers:

Reviewer #1: Thank you for inviting me to rereview this manuscript following author comments. I appreciate the depth and approach to address the comments by the authors throughout.

The authors have used arguments developed from mediation analysis to justify their approach of not including gestational age at birth within their initial protocol. I do understand that the authors propose that gestational age at birth itself is likely a mediator between the exposure (ICP) and the outcome (NCD). However, the gestation of birth is determined not only by spontaneous labour onset, but also by clinician-initiated birth; including them as a single pathway risks oversimplification of quite different events, with different relationships both to the exposure (ICP) and, potentially, the outcome (although this is more speculative). Particularly for clinician-initiated birth, the opportunity in this approach of analysis would be to identify a modifiable mediator (and potentially give evidence to support or refute early birth for affected patients). Similarly, I dispute that gestational age of birth is always determined after ICP onset - in clinical practice, iatrogenic birth (e.g. by caesarean section) may be planned before ICP diagnosis and irrespective of this (e.g. in patients with other obstetric indications for early birth) - whilst I don't think that this needs including within this study, it reflects the number of assumptions that have to be made within such a study approach, and supports my recommendation of need to adjust for this (which you have done). In reviewing the mediation analysis, I should emphasise that, whilst I am familiar with the approach, I am by no means an expert in this methodology, so my ability to review that aspect was limited. I have based my understanding upon this BMJ paper (Understanding how health interventions or exposures produce their effects using mediation analysis (bmj.com)) - the editors may wish additional expert review from an epidemiologist if they require additional insight upon this methodology. 

As such, I do really appreciate the inclusion of the new supplementary table including multiple methods of analysis accounting for gestational age at birth (and would prefer it to be included in the main document - I appreciate that will be more of an editorial decision due to this being post hoc, and with the limitation of space). I still interpret the findings with demonstrated increased risk of NCD and specifically ADHD but do not agree that we can conclude increased risk of autism - I appreciate that the authors have modified their wording to reflect this (even if we disagree a little on the interpretation!).

As with all good research, I think that new evidence often raises more questions; I appreciate that the use of mediation analysis requires readers to have an understanding of epidemiological approaches with which clinicians providing direct care may be less familiar, and I anticipate that the findings may generate some useful further discussions within the field.

Reviewer #2: Thanks for the revised manuscript and responses to my original review. 

I agree with the authors that gestational age is inappropriate for the main analyses as changes in gestational age are after any impact of cholestasis, so it can't really be considered a confounder. The analysis including gestational age hints that maybe this is a mediator, but I think this is appropriately left in the appendix. 

The inclusion of the sensitivity analysis of different approaches to missing data strengthens the manuscript - given the results are very similar with MI and IVP weights this indicates that MNAR is a reasonable assumption in this case. 

 One of the reviewers has legitimate concerns around the findings of this study causing distress to parents. This revised manuscript does make the limitations of the study design clear (i.e. observational) as clear as possible. 

Reviewer #4: Thank you for the response and adjustments. I have no further questions.

[LINK]

---

## [Editor Report · Decision Letter 3]

6 Dec 2023

Dear Dr Ahlqvist, 

On behalf of my colleagues and the Academic Editor, Prof. Gordon Smith, I am pleased to inform you that we have agreed to publish your manuscript "Maternal intrahepatic cholestasis of pregnancy and neurodevelopmental conditions in offspring: a population-based cohort study of two million Swedish children" (PMEDICINE-D-23-01940R3) in PLOS Medicine.

PRESS

Sincerely, 

Louise Gaynor-Brook, MBBS PhD 

lgaynor@plos.org